

# Selective enhancement of attentional networks in college table tennis athletes: a preliminary investigation

Biye Wang, Wei Guo and Chenglin Zhou

Department of Sport Psychology, School of Kinesiology, Shanghai University of Sport, Shanghai, China

## ABSTRACT

The purpose of the study was to investigate the characteristics of the attentional network in college table tennis athletes. A total of 65 college students categorized as table tennis athlete group or non-athlete group participated in the study. All participants completed the attentional network test (ANT) which measured the alerting, orienting and executive control networks. The results showed a significant difference between the athlete and non-athlete group for executive control network ($p < 0.01$), while no differences were observed for alerting ($p > 0.05$) or orienting ($p > 0.05$) networks. These results combined suggest that college table tennis athletes exhibited selectively enhanced executive control of attentional networks.

## INTRODUCTION

The ability to selectively focus on the relevant information while ignoring irrelevant information is a basic function of our brain to ensure that we can interact with the environment effectively. This ability requires attention, which is a core function of cognitive system and regulates other cognitive functions such as memory and language (*Posner & Petersen, 1990*). More specifically, attention plays an important role in sports (*Williams, Davids & Williams, 1999*). Obviously, it is crucial for most athletes to choose the important information to process in an extreme short period of time in a competition context (*Allard et al., 1989*). Also it would be difficult to achieve any goals for athletes with easily disturbed attention. Thus, it is reasonable to speculate that the sport-specific attentional function may develop better in athletes, relative to non-athletes. However, it is still unclear whether the athletes may have a better general attentional function (*Voss et al., 2010*). Therefore, the present study focused on the transfer of sport-specific attentional function to general attentional function. Indeed, several studies have already focused on the possible relationship between athlete experience and general attentional function in a laboratory setting (*Enns & Richards, 1997*; *Memmert, 2009*; *Memmert, Simons & Grimme, 2009*; *Nougier et al., 1992*). However, these kind of studies yielded mixed results due to variation in laboratory attentional tasks (*Voss et al., 2010*). The attentional network test (ANT) developed by *Fan et al. (2002)* is one of the most dominant attention paradigms and seems to be appropriate for this kind of study. It is a short and simple computerized task that measures the attentional networks independently. The task was based on the

Corresponding author
Chenglin Zhou,
chenglin_600@126.com

well-known attention network theory proposed by *Petersen & Posner (2012)* and *Posner & Petersen (1990)*. According to this theory, the attention system could be divided into three different networks: alerting network, orienting network and executive control network. Each of them representing a set of certain attentional functions and little overlap between the three networks was revealed by a neuroimaging analysis (*Fan et al., 2005*). The alerting network is related to maintenance of certain levels of arousal and sustained vigilance, the orienting network allows selection of information from multiple sensory inputs, and the executive control network is related to the ability to monitor and resolve conflict (*Petersen & Posner, 2012*; *Posner & Petersen, 1990*).

Although few studies have explored the three attentional networks of athletes in one experiment using the ANT, there is some evidence showing the characteristics of alerting, orientation or executive control in athletes in different studies. The alerting and orientation ability of athletes is mainly measured by the spatial cueing paradigm (*Posner & Fan, 2008*). For example, *Enns & Richards (1997)* used different cue-target intervals to investigate the alerting effect. The results revealed that athletes sustained a high level of alertness over the longest cue-target interval (*Enns & Richards, 1997*). *Cereatti et al. (2009)* observed athletes outperform non-athletes on the voluntary orientation of attention (*Cereatti et al., 2009*). Studies have also demonstrated athletes to exhibit higher proficiency on tasks testing executive function (*Jacobson & Matthaeus, 2014*; *Vestberg et al., 2012*; *Verburgh et al., 2014*). For example, *Jacobson & Matthaeus (2014)* revealed that athletes performed better than non-athletes on a problem solving as well as an inhibition task, suggesting that athletes achieved better executive control ability.

This study was designed to investigate the association between sports training experiences and the modulation of attentional network functions. It could, to some extent, answer a basic question in brain plasticity research on whether an individual's experience can affect the attentional process. Athletes are one of the most suitable models to investigate this question because of their unique experience. Compared with non-athletes, most of them trained with larger amount regularly for several years. Although it seems that previous studies have already focused on this topic for decades, the present study and these studies differ in many aspects. Firstly, athletes from one of the typical open-skilled sports, table tennis, served as the athlete group in this study. Previous studies mainly explored the attentional function of athletes from closed-skill sports (e.g., swimming, running) rather than athletes from open-skilled sports (e.g., tennis, table tennis) (*Voss et al., 2010*). Compared to closed-skill sports, open-skill sports require individuals to invest higher cognitive effort in the unpredictable environment which may serve as cognitive training to enhance the attention skill (*Tang & Posner, 2009*). It has been shown that open-skill athletes are more flexible in visual attention, decision making, inhibition, and working memory, compared to closed-skill athletes (*Voss et al., 2010*; *Wang et al., 2013*; *Heppe et al., 2016*). Secondly, the attentional network test (ANT) was adopted in this study to evaluate the efficiencies of the three attention networks in one experiment, it is more efficient than the battery of attention test mainly used in previous studies because the ANT requires only about 15 min to complete, and there are very little overlaps among the three networks. It has been widely used in certain

clinical populations, however few studies have investigated the differences between athletes and non-athletes on the ANT. To the best of our knowledge, this is the first study to investigate the characteristics of table tennis athlete's attentional networks with the ANT.

The present study aimed to investigate the characteristics of the attentional network in college table tennis athletes using the ANT. Although previous studies have indicated that chronic exercise (*Pérez et al., 2013*) and acute exercise (*Chang et al., 2015*) improve the performance on ANT in non-athletes, this was the first study to our knowledge to adopt table tennis athletes as the participants. There are three reasons for choosing table tennis athletes as the participants. Firstly, table tennis is one of the fastest ball sports and the response window dictated by the ball speed is very brief. The table tennis athletes have to use advanced cues to decide what response is required as soon as possible (*Padulo et al., 2015*), and therefore, they would develop superior alerting and orienting ability. Secondly, table tennis is a highly developed tactical skill, involving creativity, concentration, competitiveness, apprehension, self-regulation, and will power (*Raab, Masters & Maxwell, 2005*). Table tennis athletes compete in a dynamically changing, unpredictable, and externally-paced environment which may lead to better executive control ability. Thirdly, table tennis is one of the most popular sports in China. Table tennis athletes are trained systematically and have a high competition level, so they are the perfect samples to investigate the relationship between athlete training experience and attentional function. Based on the results of previews studies which focused on the three networks of attention separately, it was hypothesized that athletes would perform better on the alerting, orientation and executive network than non-athletes.

## METHOD

### Participants

A total of 65 individuals categorized as athletes or non-athletes participated in the study. They were recruited through advertisements posted in the campus of Shanghai University of Sport. The athlete group was composed of 31 table tennis players (mean age = 21.9, ranging from 19 to 25, 11 females) whom satisfied all of the following criteria: (1) had 5 or more years of professional training experience; (2) qualified as the National Player at Second Grade or above; (3) trained more than three times a week in the last two years; and (4) trained for two or more hours each time. The non-athlete group was composed of 35 students (mean age = 21.9, ranging from 19 to 25, 14 females) majoring in psychology or kinesiology. The non-athlete group matched the athlete group in age and education, but they had no experience of playing table tennis, nor any experience of athlete training. The non-athlete group had a moderate physical activity level which was measured by the Taiwan version of the International Physical Activity Questionnaire (IPAQ) (*Liou et al., 2008*). All the participants were right-handed and had normal or corrected to normal visual acuity. No individuals reported having a history of neurological or psychiatric disorder. Written informed consent was obtained from each participant prior to the study. All participants received a payment of approximately $10 for taking part in the experiment. Table 1 shows the main characteristics of the subjects. This study was approved by the Ethics Committee of the Shanghai University of Sport (No. 2015014).

**Table 1 The main characteristics of the subjects in different groups.**

|  | Athlete group($n = 31$) | Non-athlete group($n = 34$) |
|---|---|---|
| Female | 11 | 14 |
| Age (yr) | $21.90 \pm 1.72$ | $21.91 \pm 1.80$ |
| Height (cm) | $1.73 \pm 0.08$ | $1.69 \pm 0.10$ |
| Weight (kg) | $65.18 \pm 9.38$ | $61.13 \pm 9.67$ |
| BMI (kg/m$^2$) | $21.69 \pm 1.72$ | $21.32 \pm 1.95$ |
| IPAQ(METs/week) |  |  |
| *Vigorous(METs/week)* | $3587.09 \pm 2372.72$ | $2037.65 \pm 5109.58$ |
| *Moderate(METs/week)* | $1597.42 \pm 1659.15$ | $927.06 \pm 1386.74$ |
| *Walking(METs/week)* | $1448.47 \pm 1763.65$ | $1297.68 \pm 1261.23$ |
| *Overall(METs/week)* | $6632.99 \pm 3808.16$ | $4262.38 \pm 5229.69$[*] |
| Reaction time (ms) | $475.88 \pm 48.43$ | $488.45 \pm 34.94$ |
| Accuracy (%) | $97.93 \pm 1.93$ | $98.05 \pm 1.68$ |

**Notes.**

[*]$p < 0.05$.

BMI, body mass index; IPAQ, International Physical Activity Questionnaire; METs, metabolic equivalents.

## Attention network test

Attention network test (ANT) was designed to assess the function of the three different attention networks (*Fan et al., 2002*). A fixation cross was presented in the center of a computer screen at the onset of each trial. After a random interval of 400–1,600 ms, cues would present in one of the four possible conditions: no cue, center cue (the fixation cross was replaced by an asterisk), double cue (two asterisks were respectively displayed above and below the fixation cross), or spatial cue (an asterisk were displayed either above or below the fixation cross). The cues remained visible for 100 ms. The presentation of asterisks provided temporal information about the appearance of target stimuli. The asterisk in the spatial cue condition provided additional information about the location of target stimuli. The spatial cues were always valid. The fixation cross was displayed alone for 400 ms after the disappearance of cue. Then a target stimulus was presented above or below the fixation cross according to the indication of the previous cue. The target stimulus consisted of five horizontally arranged arrows or lines. Participants were required to press the corresponding key to indicate the direction of the central target arrow. The other four arrows or lines served as flankers in the task with three possible conditions: congruent condition (arrows pointed in the same direction as the central arrow), incongruent condition (arrows pointed in the opposite direction of the central arrow), or neutral condition (lines with no direction information). The target stimulus remained on the screen until the participant responded or for 1,700 ms if no answer was given.

The participants were instructed to concentrate on the fixation cross throughout the task. A numeric keyboard was placed in front of the participant and the participant was required to lightly put his left hand index finger on key "1" and right hand index finger on key "3." Once target stimuli were presented, participants were instructed to respond as fast and accurately as possible by pressing the key "1" for left directed central target arrow and pressing the key "3" when the direction was right.

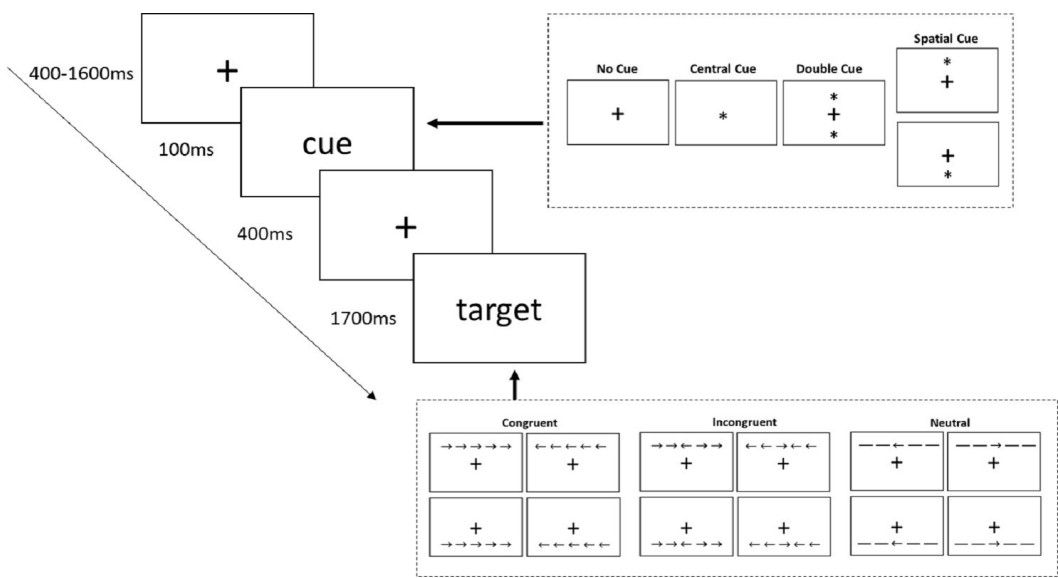

**Figure 1 Stimuli and experimental paradigm of Attention Network Test (ANT).**

Four blocks were included in this test. Each block contained 48 trials based on the combination of four cues conditions (no cue, center cue, double cue, and spatial cue), three flankers' conditions (congruent, incongruent, and neutral), two directions (left or right directed target arrow) and two locations (target displayed above or below the fixation cross). Each trial was presented only once in a block. The stimuli were presented and the data were recorded using Psychtoolbox (*Brainard, 1997*) (see Fig. 1).

The three components of attentional network were computed as follows: no cue RTs versus double cue RTs for alerting, central cue RTs versus spatial cue RTs for orienting and congruent flankers RTs versus incongruent flankers RTs for executive network.

## Procedure

As a requirement of the advertisements, all the participants had to contact the researchers by telephone first. A survey about the demographic data of participants was conducted during the call. Athlete participants were further asked about their training experiences. Participants who met the criteria (see 'Participants') were invited to our laboratory on another day to participate in the experiment. They were instructed to abstain from alcohol for 24 h and from caffeine-containing substances for 12 h before the experiment.

After arriving at the laboratory, participants were asked to sign an informed consent form and were assessed by the Taiwan version of the International Physical Activity Questionnaire (IPAQ). Then the purpose of the study and the instruction of ANT were introduced to them in written form. After participants reported understanding the instructions, they performed the ANT task individually in a dimly lit and quiet room. At first, they had to perform a practice block with 24 random trials. If their response accuracy reached 80%, they could perform their next four experimental blocks of 48 trials in each; otherwise, they would perform another practice block until their accuracy reached 80%. Participants were allowed to rest between each block, and they could start the next block by pressing any

keys once they felt adequately rested. Completing the whole task required about 17 min, including both practice and experimental blocks.

## Design and statistical analysis

A mixed factors design was adopted in the study. The athlete and non-athlete group was a between-subjects variable, the cue type (no cue, central cue, double cue and spatial cue) and flankers type (neutral, congruent, incongruent) were within-subject variables. The dependent variables were response times (RTs) and accuracy rates. They were analyzed with a 2 (group) × 4 (cue type) × 3 (flanker type) mixed-design ANOVA.

A $t$-test between athlete and non-athlete groups was carried out in order to explore the effect of athlete experience on each component of attentional network.

# RESULTS

## Participant characteristics

No significant differences were observed in age ($F_{(1,63)} = 0.00, p = 0.98$), height ($F_{(1,63)} = 3.29, p = 0.07$), weight ($F_{(1,63)} = 2.92, p = 0.09$), BMI ($F_{(1,63)} = 0.64, p = 0.43$), average reaction time ($F_{(1,63)} = 1.45, p = 0.23$) and accuracy rate ($F_{(1,63)} = 0.07, p = 0.79$), and as expected, a significant difference was observed in physical activity level (overall score on IPAQ) ($F_{(1,63)} = 4.29, p < 0.05$) of the two groups (see Table 1).

## Mean RTs

For the RTs analysis, an outlier correction was done by excluding the trials which were 3 standard deviations from the mean for each flanker condition (congruent, incongruent and neutral) individually. The method of outlier correction was suggested by one reviewer. The wrong trials were also excluded, and the proportion of excluded data was 1.1%. Results showed a significant main effect of cue type ($F_{(3,189)} = 138.82, p < 0.01, \eta_p^2 = 0.70$), the RTs were the longest in the no cue condition, and the shortest in the spatial cue condition. A significant main effect also observed in flanker type ($F_{(2,126)} = 318.31, p < 0.01, \eta_p^2 = 0.84$). The RTs were longer in the incongruent condition than in the congruent or neutral condition. Furthermore, there were significant interactions between flanker type and cue type ($F_{(6,378)} = 7.90, p < 0.01, \eta_p^2 = 0.11$), group and flanker type ($F_{(2,126)} = 4.68, p < 0.01, \eta_p^2 = 0.7$). The interaction contrasts for flanker type and cue type revealed significant differences between the congruent and incongruent conditions, incongruent conditions and neutral conditions under all cue conditions, no significant differences were observed between congruent and neutral conditions under all cue conditions. The interaction contrast for the group and flanker type revealed significant differences between the groups under incongruent condition, no significant differences were observed between the groups under congruent and neutral conditions. There were no significant main effect of group ($F_{(1,63)} = 1.45, p = 0.23, \eta_p^2 = 0.02$), group × cue type ($F_{(3,189)} = 0.63, p = 0.60, \eta_p^2 = 0.01$) or group × cue type × flanker type ($F_{(6,378)} = 0.41, p = 0.87, \eta_p^2 = 0.00$) interaction. The description data of the mean RTs and standard deviations of athlete and non-athlete group according to the cue and flanker type are shown in Table 2.

**Table 2  Mean RTs (ms) and standard deviations of athlete and non-athlete group according to cue and flanker type.**

|  | Congruent | | Incongruent | | Neutral | |
|---|---|---|---|---|---|---|
|  | Athlete | Non-athlete | Athlete | Non-athlete | Athlete | Non-athlete |
| No cue | 477.3 ± 50.8 | 486.1 ± 36.2 | 532.6 ± 55.2 | 554.0 ± 49.4 | 485.6 ± 55.3 | 492.0 ± 42.9 |
| Central cue | 455.8 ± 49.0 | 459.8 ± 38.6 | 520.2 ± 63.5 | 537.3 ± 43.1 | 458.4 ± 49.4 | 465.6 ± 40.7 |
| Double cue | 459.7 ± 49.7 | 465.0 ± 41.1 | 519.6 ± 56.3 | 542.0 ± 43.5 | 457.7 ± 48.7 | 467.4 ± 43.2 |
| Spatial cue | 434.0 ± 52.0 | 446.2 ± 36.5 | 472.3 ± 51.6 | 501.4 ± 43.8 | 435.9 ± 48.1 | 443.4 ± 36.9 |

**Table 3  Mean accuracy (%) and standard deviations of athlete and non-athlete group according to cue and flanker type.**

|  | Congruent | | Incongruent | | Neutral | |
|---|---|---|---|---|---|---|
|  | Athlete | Non-athlete | Athlete | Non-athlete | Athlete | Non-athlete |
| No cue | 96.4 ± 4.5 | 96.9 ± 5.6 | 94.8 ± 4.3 | 93.6 ± 6.1 | 99.0 ± 2.8 | 99.3 ± 2.0 |
| Central cue | 98.2 ± 2.9 | 97.8 ± 3.7 | 91.9 ± 8.9 | 94.5 ± 6.7 | 98.6 ± 2.7 | 99.1 ± 2.2 |
| Double cue | 99.0 ± 2.3 | 97.2 ± 4.4 | 94.4 ± 6.7 | 95.8 ± 5.0 | 98.8 ± 2.5 | 98.3 ± 2.8 |
| Spatial cue | 99.4 ± 1.9 | 98.0 ± 3.7 | 96.8 ± 4.8 | 97.6 ± 5.3 | 99.2 ± 2.7 | 99.3 ± 2.0 |

## Accuracy

For the accuracy analysis, significant main effects of cue type ($F_{(3,189)} = 7.89, p < 0.01, \eta_p^2 = 0.11$), and flanker type ($F_{(2,126)} = 39.9, p < 0.01, \eta_p^2 = 0.39$) were revealed. Furthermore, there were significant interactions between flanker type and cue type ($F_{(6,378)} = 4.10, p < 0.01, \eta_p^2 = 0.6$). Interaction contrast revealed significant differences between the congruent and incongruent conditions, incongruent conditions and neutral conditions under all cue conditions, no significant differences were observed between congruent and neutral conditions under all cue conditions. There were no significant main effect of group ($F_{(1,63)} = 0.03, p = 0.87, \eta_p^2 = 0.00$), group and flanker type ($F_{(2,126)} = 1.59, p = 0.21, \eta_p^2 = 0.3$), group × cue type ($F_{(3,189)} = 0.88, p = 0.45, \eta_p^2 = 0.01$) or group × cue type × flanker type ($F_{(6,378)} = 1.59, p = 0.15, \eta_p^2 = 0.2$) interaction. The descriptive data of the mean accuracy and standard deviations of athlete and non-athlete group according to the cue and flanker type are shown in Table 3.

## Differences of athletes and non-athletes on the 3 components of attentional network

Independent samples $t$-tests were carried out for each component of the attentional system (alerting, orienting and executive networks). Results showed a significant difference between athlete and non-athlete group on executive network ($t_{(63)} = 2.36, p = 0.02$), while no differences were observed on alerting ($t_{(63)} = -0.05, p = 0.96$) or orientation ($t_{(63)} = -1.32, p = 0.19$) networks (see Fig. 2).

## DISCUSSION

The aim of the present study was to investigate the relationship between sports training experience and the attentional network using the ANT. Our results showed that the athlete

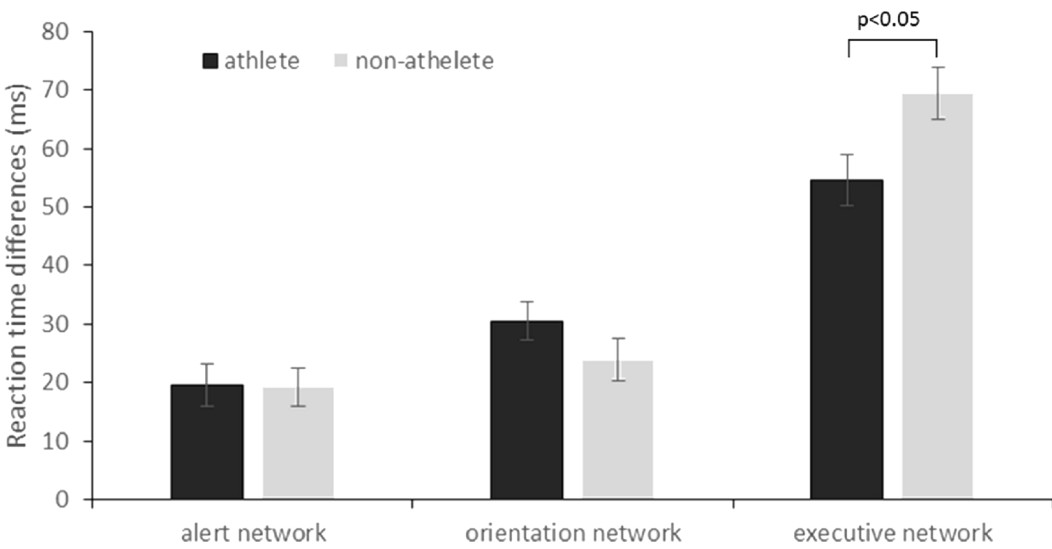

**Figure 2** **Reaction time differences that reflect the efficiency of the three attentional networks of athlete and non-athlete group (mean ± SE).** The smaller differences on executive network and the larger differences on alerting and orientation network indicate a better function.

group received a higher score than the non-athlete group on the executive network component, which is consistent with previous findings that have confirmed a positive correlation between executive control and athletic ability (*Jacobson & Matthaeus, 2014*; *Vestberg et al., 2012*). A possible reason for the superior executive network function of athletes may be mainly due to the cognitive benefit of physical activity. Also, it has been proposed that exercises performed in the cognitively challenged environment are more effective to induce neural and cognitive benefits than exercise alone (*Fabel et al., 2009*). Table tennis athletes train and compete in the kind of enrichment environment that includes both physical and mental challenges. However, the present study cannot infer a causal relationship between athletic experience and attentional network function. It is possible that individuals who develop strong executive control skills are more likely to become athletes. *Vestberg et al. (2012)* suggest that individuals with high executive control ability become athletes more often and the ability further improved with training. It is speculated that the observed differences in attentional network may, at least in part, result from athletic participation.

The alerting and orientation of attention are especially important for athletes because they have to keep alerted all the time and orientate their attention quickly to the relevant information in the sporting context. However, the efficiency of the alerting and orientation networks tested by ANT did not differ in athletes and non-athletes in the present study. These results were inconsistent with previous findings, which have revealed that athletes practicing open-skilled sports showed superior ability on the alerting and voluntary orientation of attention than their counterbalanced controlled non-athlete group (*Enns & Richards, 1997*; *Nougier et al., 1992*). Both of these studies measured the alerting effect by testing more than one stimulus onset asynchrony (SOA) between cue and target, and the orienting effect was measured by comparing the reaction time difference between target stimuli at attended and unattended locations. However, the efficiency of alerting

and orientation network tested by ANT were equivalent in athletes and non-athletes in the present study. This is consistent with the meta-analysis by *Voss et al. (2010)*. They found the effects of athlete experience were small and not statistically significant ($g = .17; p > .05$) in attentional cuing paradigm which is similar to the alerting and orienting network tests of the ANT in the present study. A possible reason for the inconsistency may be mainly due to the different experimental paradigms. The ANT used in this study is a relative simple task, and the response times for the measurement of orienting might have been affected by a ceiling effect. Also, the participants in the non-athlete group seemed to participate in regular physical exercise which could improve their cognitive function (*Voss et al., 2011*).

The selective enhancement of the executive control network in athletes is similar to previous studies focused on the effect of chronic exercise or acute exercise on alerting, orientation, and executive control using a similar version of the ANT. *Pérez et al. (2013)* found a difference between active and passive participants on the executive network while no differences were observed on the alerting and orientation network. Along the same line, *Chang et al. (2015)* found that rather than eliciting general improvement, a single bout of acute exercise selectively enhanced executive control of attention.

The present study also revealed a significant interaction between flanker type and cue type, suggesting that the orientation cue was most effective when conflict resolution was required, while the alerting cue failed to increase the efficiency of executive control. It mirrored the pattern of interactions obtained in an earlier study with adults using the ANT (*Fan et al., 2002*). The interaction between group and flanker type was consistent with the result that athletes were more efficient on the executive network.

Some limitations existed in the present study. Firstly, the cross-sectional design revealed a possible relationship between athletic experience and the attentional network, but it can hardly conclude a causal relationship. Longitudinal studies are needed in the future. Further, this design did not allow for deep exploration of the cause of selective enhancement of executive control of attention. Also, all the athlete participants in the study were qualified as the National Player at Second Grade. Athletes in different sport levels (e.g., elite and novice) should be enrolled in a future study to specify the relationship between attentional network and expertise in sports.

## CONCLUSION

In conclusion, college table tennis athletes exhibited selective enhancement of execution control of attentional networks while no differences between athletes and non-athletes were observed in the alerting and orientation networks. It suggests the existence of certain association between sports training experiences and the modulation of the executive control network.

### Funding
This work was supported by the National Natural Science Foundation of China (grant numbers 31571151); and the Graduate Education Innovation Project of Shanghai

University of Sport (grant number yjscx2016004). The funders had no role in study design, data collection and analysis, decision to publish, or preparation of the manuscript.

### Grant Disclosures

The following grant information was disclosed by the authors:
National Natural Science Foundation of China: 31571151.
Graduate Education Innovation Project of Shanghai University of Sport: yjscx2016004.

### Competing Interests

The authors declare there are no competing interests.

### Author Contributions

- Biye Wang conceived and designed the experiments, performed the experiments, analyzed the data, contributed reagents/materials/analysis tools, wrote the paper, reviewed drafts of the paper.
- Wei Guo conceived and designed the experiments, performed the experiments, analyzed the data, contributed reagents/materials/analysis tools, wrote the paper, prepared figures and/or tables.
- Chenglin Zhou conceived and designed the experiments, reviewed drafts of the paper.

### Human Ethics

The following information was supplied relating to ethical approvals (i.e., approving body and any reference numbers):

This study was approved by the Ethics Committee of the Shanghai University of Sport (No. 2015014).

### Data Availability

The raw data has been supplied as a Supplementary File.

### Supplemental Information

Supplemental information for this article can be found online at http://dx.doi.org/10.7717/peerj.2762#supplemental-information.

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
