# Peer review of "Selective enhancement of attentional networks in college table tennis athletes: a preliminary investigation"

_PeerJ, doi:10.7717/peerj.2762_

## Round 0.1 · original submission · Major Revisions

The manuscript is very interesting for PeerJ, and can contribute to the development of knowledge in the field. However two out of three reviewers highlighted many important issue that need your attention. Please see the specific comments below and address all the comments.

·

Basic reporting

The article is written well and flows. There are a few typos and awkward sentences , please see annotated PDF file for specific suggestions and edits. The introduction and literature review sections are developed well. However, perhaps a bit more needs to be explained why you chose to study general attention and not domain specific attention which seems to be more relevant to attention acquired through playing sports. Related to this, you need to clarify why you conducted the study, why is it interesting to examine this? Figures and tables are presented clearly

Experimental design

The purpose of the study is clear. The research question and hypotheses are stated. However, they need to be a bit more detailed. The authors clearly state the gap the article fills, specifically looking at the differences in attention functions between athletes participating in an open-skill sport (table tennis) compared to non-athletes. The method section includes most of the relevant information. However, I did not see information about the gender of the participants. In addition, maybe some descriptive data on the table tennis athletes experience could be presented.

Validity of the findings

The discussion section should be organized a bit better. Specifically, the first paragraph should be moved to either the introduction or method section (see annotated PDF for specific details). Furthermore, a few plausible explanations should be presented on why only the executive control network was different between the groups (and not orienting and alerting). Good limitations.

Additional comments

The article is written well, flows and is clear. The topic is relevant to the PeerJ journal, specifically to the area of the relationship between cognition (i.e., attention) and sport (i.e., open skill – table tennis). The authors’ examined general attention skill differences (using the ANT) between table tennis players and non-athletes. In my opinion, only minor revisions are needed. Please see comments and annotated PDF file for specific suggestions and feedback.

Reviewer 2 ·

Basic reporting

The presented paper addresses an important question in the field. The manuscript would benefit from professional editing to improve the readability and reduce the number of careless mistakes, i. e. typos, reference list and citations within the text.
The authors attached a data file called raw data. However, the file consists of (arithmetic?) means for every participant. It would be praiseworthy if original logfiles were uploaded.
The introduction provides a good overview on the current state of research. Articles that could be included in the introduction are:
Heppe, H., Kohler, A., Fleddermann, M. T., & Zentgraf, K. (2016). The relationship between expertise in sports, visuospatial and basic cognitive skills. Frontiers in Psychology, 7, 904.
Tang, Y. Y., & Posner, M. I. (2009). Attention training and attention state training. Trends in cognitive sciences, 13(5), 222-227.
Verburgh, L., Scherder, E. J., van Lange, P. A., & Oosterlaan, J. (2014). Executive functioning in highly talented soccer players. PloS one, 9(3), e91254.
The authors cited the meta-analysis by Voss et al. (2010) and concluded that there are mixed findings. Voss et al. subdivided attentional skills into attentional cuing, processing speed and varied attention paradigms. The paradigm in this study can be categorized into attentional cuing. Voss et al. found a smaller effect size for attentional cuing (g < .2; p > .05) compared to other cognitive measures. The authors could take a look at included studies by Voss et al. which were using a similar paradigm as they did and find reasons why they found a larger effect size (other level of expertise, other control group, other sports they athletes did…?).
Line 222-226: In the Discussion, the authors mention a possible ceiling effect for the measurement of orienting. This could also have implications for the interpretation of the executive network results, because the calculated score contains this measure.

Experimental design

Overall, it is appreciated that the method section is pretty detailed, so the research seems to be replicable with given information and the task should be understood. To further improve the reproducibility, the Matlab code could be shared.
Two recommentations on chapter 2.1 (Participants):
1. Giving information on gender, because it is only noted in Table 1.
2. Giving further information on the control group and describing them as detailed as possible. Since some of them are fairly active, it is interesting if they participate in interactive or other sports. Otherwise, it could be recommended to include this into the paragraph on limitations of the study.
In 2.2, the authors describe the attention test. It could be considered to include the paragraph in 2.4 on the calculations on each component of attentional network here. In the explanation of the computation of values representing each network, it would be helpful to the reader to use identical names for the executive network (and not “conflict resolution”). There is one open question to the random interval of 400 to 1600 ms: Was it absolutely random or varied in 100 ms steps?
In 2.4, the formatting is not correct yet (heading, as in 3.x).
The conclusion is not optimal at the moment since it only repeats the main message yet. It should be extensively reworked.

Validity of the findings

The results sections starts with a detailed description of the participants in both groups (Table 1). The authors compare every aspect (age, height…) with a t-test. This seems not necessary. If the authors decide to stay with it, they could consider reporting effect sizes and confidence intervals and results should be reported in APA format, because the p-value does not help a lot in this case. Mean RT over all trials and accuracy are not compared. However, it can be questionable whether a mean over all trials is a valid measure for a group comparison. Did the authors want to show that average choice response time did not differ between both groups? If yes, is there a better measure (like neutral condition, spatial cue)?
In 3.2 (Mean RTs), trials (typo: trails, line 148) more than 3 SDs from the individual mean were classified as outliers. It is laudable that no fixed limits were used. However, this method seems to be not appropriate to the data, because response times differ between conditions. Therefore it can be assumed that more trials from the incongruent/ no cue condition were excluded compared to other conditions. The authors should do the outlier correction for each condition, especially for the incongruent condition. The reader should be informed about the proportion of excluded data.
In 3.4, independent samples t-tests of the 3 components of attentional network are conducted. These are new measured built from scores analyzed in the previous analyses. Before doing the post-hoc tests, an ANOVA including all 3 components (with present data, the eta² of interaction*group is .11 – this part strengthens the analysis). Furthermore, p-values do not tell the reader about the size of an effect, so providing a measure of effect size is recommended. Post-hoc tests which explain notewothy interactions (e. g., group and flanker type) seem to be missing. The reader could be interested why these interactions occur without having to look into the table.
Line 71-76: It would be interesting how the athlete group was composed: How many players were qualified as 1st and 2nd grade? Is there a difference within the athletes?
Line 79-83: Since all participants were right-handed (it would be interesting to know how this was evaluated), it is interesting to know whether there was a laterality difference in both groups in response times.

Additional comments

Figure 2 should be be comprehensible when detached from the text. Therefore it should be added that shorter differences indicate a better performance.
It seems like the error bars in the figure represent standard errors. Since SDs and no SEs are reported in other parts of the paper, the Figure should contain SDs and no SEs (without mentioning them).

Reviewer 3 ·

Basic reporting

The intro is not clear in some instances and I would clarify better the last paragraph:
“Although previous study have indicated that chronic exercise and acute exercise improve the performance on ANT in non-athletes (Chang, Pesce, Chiang, Kuo, & Fong, 2015; Pérez, Padilla, Parmentier, & Andrés, 2013), this was the first study to our knowledge to adopt athletes as the participants.” Please explain and motivate better since is not clear whether the learning effect was present while learning the ANT test or practicing some other perceptual motor cognitive tasks.
Following: “It was hypothesized that athletes would perform better on the alerting, orientation and executive network than non-athletes”. This phrase needs to be sustained again on why this should be the case.

Experimental design

The non-athletes group is a non-homogeneous (psy and kines) group and in addition I imagine that some of the kines students are athletes even though they are not professional. It has been shown in several instances that athletes having an intermediate level of sport experience are “able to locate their attention” similarly as elite athletes.
Why is defined in the practice trial a percentage of 80% of level of accuracy?
Please clarify better :“A t-test carried out in order to explore the effect of athlete experience on each component of attentional network” a t-test between groups for each attentional components? A t-test within group and just for athletes? This is not clear.

Validity of the findings

In the results explain or report reference about this: “For the RTs analysis, the wrong trails or the trails which were 3 standard deviations from the individual mean were excluded”
Please explain the interaction effect: Furthermore, there were significant interactions between flanker type and cue type (F (6,378) =6.95, p <0.01, ηp2 =0.10), group and flanker type (F(2,126) =5.90, p <0.01, ηp2 =0.09).
And in the accuracy results as well:
“Furthermore, there were significant interactions between flanker type and cue type (F (6,378) =4.80, p <0.01, ηp2 =0.07)”.
Line 173: independent samples t-test for comparing the two groups ….for each components …
Here I do not understand since the ANOVA already performed contains this information. Or you have to explain better how the three attentional networks were derived. In the discussion please report biblio here: ”Previous studies mainly explored the attentional function of athletes from closed-skill sports (e.g. swimming, running) rather than athletes from open-skilled sports (e.g. tennis, table tennis).”
“It is speculated that the observed differences in attentional network may, at least in part, result from athletic participation.” It is also possible that airplane pilots or simply video games players may show the same ability? If so it is not just related to sport practice or to sport specificity.
From line 223 to line 226: I do agree with the authors and I believe that this is the major weakness of this experiment. In too many instances it has been shown that task specificity is fundamental to show abilities that are sport-specific.

Additional comments

The experiment is aimed at revealing the difference between table tennis players and a control group by applying a computerized attentional test.
The main limitations of the work are:
The lack of a “convincing” control group (new data collection is required)
A small number of participants tested
A lack of a clear explanation of the significant as well as non-significant differences found as results as I explained in more details above.

---

## Round 0.2 · Major Revisions

The authors have addressed the suggestions and the manuscript is much better in its current format. However some major concern remain. Specifically, I agree with the second reviewer that the outlier-correction is not appropriate to the data, because the conditions differentiate in difficulty (and as a result, in RTs). Therefore, when performing an outlier-correction, it has to be done for each condition individually. Otherwise, outliers in the easier conditions are missed. What I suggest is try to apply outlier correction for each condition and then run again the mixed factors design analysis. Please read the comments of each reviewer below.

·

Basic reporting

The authors have addressed the suggestions and the manuscript is much better in its current format.
Specifically, they have added more information on the rationale of choosing a general attention test.

Experimental design

The authors have addressed the comments related to the experimental design and procedure. They could still add a table that includes information related to the criteria the participants where chosen by: training experience, level of play, training time per week, training hours per session. It is good that they used clear criteria, but it will be better if they showed that there were actual differences between the two groups on these variables.

Validity of the findings

The authors have rearranged the discussion and added a few important sections. I still think that a better explanation is needed for why the executive control network was different and not the orienting and altering. This is not common sense, as their hypothesis stated that all 3 skills should be different among groups. I would advise them to expand the explanation.

Additional comments

The authors have improved the manuscript and have addressed most of the comments. The article is much better, flows and includes important information throughout the sections. It should be accepted as is, with only minor changes, that will improve it, but that are not crucial and will leave it to the authors if they wish to make them or not. .

Reviewer 2 ·

Basic reporting

Thank you for sharing the raw data, I have just one open question: On July the 6th you had 31 participants in each group. Why did you collect data of 3 further participants for the control group one week later?

Experimental design

No comments.

Validity of the findings

Thank you for your explanation on your outlier correction method. I am still not convinced by the method, because it leads to an underestimation of the classification of "long RT" outliers in the ANT and an overerstimation of outliers in other conditions, e.g. executive network. If there are "long RT" outliers in the ANT, they are not classified correctly because the mean of all trials is higher than the mean on the ANT trials. Therefore, response times are included which are supposed to be outliers by the 3 SD outlier correction method.
Since the reported sign. difference between athletes and non-athletes is about 15 ms (Fig. 3), I think that my point of a careful outlier analysis is valid.

---

## Round 0.3 · accepted · Accept

I'm very happy that you have found the reviewer's suggestion useful, and that you revised the manuscript accordingly. In particular, I appreciated your effort in performing the new outlier correction.

Congratulation again.